# Vaccinating to Protect Others: The Role of Self-Persuasion and Empathy among Young Adults

**DOI:** 10.3390/vaccines10040553

**Published:** 2022-04-02

**Authors:** Dariusz Drążkowski, Radosław Trepanowski, Valerie Fointiat

**Affiliations:** 1Faculty of Psychology and Cognitive Science, Adam Mickiewicz University, 61-712 Poznan, Wielkopolska, Poland; radtre@amu.edu.pl; 2LPS, Aix-Marseille Université, 13621 Aix-en-Provence, France; valerie.fointiat@univ-amu.fr

**Keywords:** COVID-19 vaccination, health communication, self-persuasion, planned behavior theory, empathy

## Abstract

Direct persuasion is usually less effective than self-persuasion. As research shows that most young adults are unafraid of COVID-19, this study aimed to investigate the effectiveness of self-persuasion targeted at protecting the health of others to encourage young adults to be vaccinated against COVID-19 and examined the link between empathy and vaccination intention. We conducted two studies: Study 1 (*n* = 352) compared the effectiveness of self-persuasion targeted at others’ health versus personal health and direct persuasion in encouraging COVID-19 vaccination intention; Study 2 (*n* = 375) investigated the applicability of self-persuasion through a poster framed as an open-ended question. The theory of planned behavior-based tools were used in both studies, and structural equation modeling was conducted. Study 1 found that self-persuasion targeted at others’ health (compared to other forms of persuasion) indirectly affects vaccination intention through utility and social norm beliefs. Higher empathy, utility, social norms, and control beliefs are associated with a greater vaccination intention. Study 2 found that the poster with self-persuasion targeted at others’ health enhanced vaccination intention compared with a direct persuasion poster. Our findings demonstrate that self-persuasion targeted at others’ health can potentially increase COVID-19 vaccination uptake among young adults.

## 1. Introduction

Vaccination against COVID-19 can be considered part of the solution to overcome the ongoing pandemic [1]. While several vaccines have been developed, the effectiveness of vaccination programs depends on high uptake rates among the population. In particular, young adults (18–30 years; [2]) are less likely to be vaccinated against COVID-19 as they are less afraid of contracting the disease than their older counterparts [3,4]. Thus, it is crucial to develop evidence-based health communication and encourage young adults to be vaccinated against COVID-19. Approaches that examine persuasive techniques in tandem with motivational factors can address these challenges [5]. In this study, we investigated the effectiveness of self-persuasion targeted at protecting the health of others to encourage young adults to be vaccinated against COVID-19.

Governments across the world have been employing various strategies, such as persuasive communication [6], to promote vaccine uptake. Some studies have demonstrated that direct persuasion (persuasive arguments from external sources) may increase COVID-19 vaccination intention [7]. However, other studies have shown that attempts to encourage COVID-19 vaccination through direct persuasive messages are ineffective [8]. Direct persuasion may also lead to a boomerang effect, as it tends to trigger recipients’ psychological reactance [9]. Opposite effects of direct persuasion have also been observed in prior studies, in which people were encouraged to be vaccinated [10].

Compared to direct persuasion, a subtler persuasive technique is self-persuasion—the process of generating one’s own arguments toward a specific issue [11]. Unlike receiving arguments from external sources (as with direct persuasion), creating one’s own argument reduces the risk of psychological reactance [12,13]. Individuals creating arguments that contradict their attitudes experience cognitive dissonance, which motivates them to seek consistency in their cognition by realigning their attitudes to match the created arguments [14]. Baldwin et al. [15] found another mechanism underlying effective self-persuasion. A self-generated argument is tailored to an individual’s concerns to a greater extent than arguments from external sources. Therefore, people think more positively and evaluate their arguments more favorably than external arguments. A considerable body of research has shown that self-persuasion leads to improvements in health-related behaviors. For instance, Baldwin et al. [16] demonstrated that self-persuasion might increase parents’ intention to vaccinate their adolescent children against human papillomavirus (HPV). Moreover, Drążkowski et al. [17] revealed that self-persuasion has a significant effect on one’s moral obligation to socially isolate during the COVID-19 pandemic, and through it, on self-isolation intention.

One of the major challenges in the fight against the COVID-19 pandemic is encouraging young adults to take the vaccine to increase the chances of populations achieving herd immunity [3,4]. To date, a number of motivational barriers to vaccination against COVID-19 have been identified among young people, including belief in conspiracy theories [18] and a lack of confidence in the safety, authenticity and efficacy of vaccination [19,20]. Young adults are less concerned about COVID-19 than older adults [21,22], as this disease does not pose a great risk to young adults’ health [23]. Thus, for young adults, arguments aimed at vaccine uptake to protect personal health (e.g., “vaccination will protect you from severe COVID-19 infection”) may be less convincing than arguments aimed at protecting the health of others (e.g., “vaccination will protect elderly and the sick from severe COVID-19 infection”). Thus, among young adults, targeting self-persuasion to protect the health of others may be more influential in improving the COVID-19 vaccination rate.

Recent research shows that willingness to protect others is an important motivation for vaccination behavior [24] and that concern for others increases the intention to get vaccinated against COVID-19 [25]. Since concern for protecting the health of others is an important motivation to be vaccinated against COVID-19, it is reasonable to expect that highly empathetic individuals, who are more concerned about the welfare of others [26], are more likely to be vaccinated than low-empathetic individuals. Empathy is defined as a set of cognitive and emotional constructs linked by responsiveness to others’ concerns [27]. One study demonstrated that willingness to be vaccinated against COVID-19 increases as empathy increases [28]. Similarly, another study found that empathy motivates individuals to practice physical distancing and mask-wearing during the COVID-19 pandemic [29]. Pfattheicher et al. [28] also suggest that promoting empathy can increase the intention to vaccinate against COVID-19, as vaccination can be conceptualized as a prosocial act that helps protect vulnerable individuals.

Identifying pathways mediating the effect of self-persuasion targeted at protecting the health of others to be vaccinated may help understand the mechanism underlying this type of persuasive communication. The theory of planned behavior (TPB; [30]) is particularly relevant to studies on vaccination intention, as it helps identify determinants that guide intentional behavior. Thus, TPB might provide more insight into the motivation for COVID-19 vaccination intention. TPB has been widely applied to understand and predict the intention to vaccinate in both pre-pandemic [31] and pandemic conditions [21,32]. Intentional behavior arises from three motivational belief systems about any given behavior: utility, social norms, and control beliefs. Utility beliefs focus on the consequences of behavior and result in a positive or negative attitude toward the behavior. Social norm beliefs, which are concerned with others’ expectations of the behavior, establish and reinforce social norms. Control beliefs are perceptions about the difficulty or effort required to execute a behavior and deal with challenges, resulting in perceived behavioral control. All three belief systems indirectly affect behavior through intention.

Vaccination behavior has a potential, substantial impact on the welfare of others, as it can protect them from contracting a severe disease [1]. Therefore, it can be considered a behavioral manifestation of moral norms. An individual’s moral norms are defined as socially determined and validated values attached to behaviors [33]. Moral beliefs can work in parallel to the TPB belief systems [34]. Furthermore, moral norms significantly predict vaccination intention when controlling for all TPB components [35] and are highly related to vaccination intention against COVID-19 [21].

## 2. Study 1

This study aimed to investigate the effectiveness of self-persuasion targeted at protecting the health of others to encourage young adults to be vaccinated against COVID-19. First, based on the TPB model [30], we can suggest that young adults under self-persuasion to be vaccinated against COVID-19 to protect the health of others (e.g., sick people and older adults) will create arguments indicating that the vaccination: (1) is effective in protecting the health of others; (2) uptake will lead to positive reactions from others; (3) is morally right because it protects others; (4) easily contributes to the protection of others. Thus, we hypothesized that self-persuasion—by utility, social norms, moral norms, and control beliefs—focused on others’ health protection would have a greater influence on intention to vaccinate compared to direct persuasion or self-persuasion targeted at one’s own health.

Second, previous studies have shown a positive association between empathy and intention to vaccinate against COVID-19 [29]. As highly empathetic people are primarily focused on others’ benefits rather than their own [26], we expect that similar mechanisms may be identified in creating arguments on vaccination focused on others’ health. By demonstrating an association between empathy and intention to vaccinate against COVID-19 among young adults, we will be able to support our assumption that the protection of others is an important motivator for young adults to vaccinate. Thus, we hypothesized that an increase in empathy—regardless of the intervention—will increase utility, social and moral norms, and control beliefs, which are associated with a greater intention to vaccinate against COVID-19. 

Furthermore, we controlled for the effects of selected factors associated with the intention to vaccinate, as these factors may be related to the response to self-persuasion focused on others’ health. Prior findings have shown that men and individuals who know someone affected by COVID-19 are more willing to be vaccinated against COVID-19 than their counterparts due to lower utility, control, and social beliefs [31,36]. In contrast, although high empathy is usually associated with a greater willingness to be vaccinated against COVID-19 [29], women are, on average, more empathetic than men [37] but less willing than men to be vaccinated [21,37]. Thus, we hypothesize that, through TPB components, women are less willing to be vaccinated than men, but the links between gender and empathy suppress this relationship. Further, as shown in a previous study [21]), we hypothesized that knowing someone with COVID-19 is associated with a greater intention to vaccinate through all TPB components.

### 2.1. Materials, Participants and Methods

#### 2.1.1. Participants and Procedure

Young Polish adults (*n* = 366) participated in an online study in late September 2020, which was carried out in accordance with the Declaration of Helsinki and its amendments [38]. The survey was conducted through Microsoft Forms, and the collected data were stored on University servers. The demographic data of the participants are presented in Table 1. Participants were recruited by disseminating a short informative post about the study across multiple Facebook groups and communities, especially those targeting young adults and students. All participants provided informed consent. Fourteen responses were excluded from the analyses: three were exactly the same entries submitted by the same participant due to an application error and twelve questionnaires had missing data. Therefore, 352 responses were analyzed. The data used in this study are available for free download from the Open Science Framework (OSF) (https://osf.io/ckf2v/?view_only=42799b0d579b437c8cb926ccfddbfba0, accessed on 1 March 2022).

First, each participant was presented with brief information about the study and their rights as participants. Subsequently, they were randomly assigned to one of the three possible versions of the study. At the beginning of each version of the study, participants completed a measure of empathy, after which the experimental manipulation occurred. In the direct persuasion group (*n* = 119), participants were asked to read two arguments in favor of getting vaccinated, which were based on the WHO’s communications (e.g., *Getting vaccinated protects against contracting COVID-19. Even young people who usually are only mildly affected by COVID-19 are at risk for serious complications, such as heart failure*). Participants rated the extent to which they agreed with each argument on a seven-point scale (1 = *I do not agree at all* to 7 = *I completely agree*). In the group with self-persuasion targeted at others’ health (*n* = 123), participants were asked to create two or more arguments in favor of vaccinating to protect others’ health (e.g., *If I don’t vaccinate, my mother could get sick)*. In the second experimental group with self-persuasion targeted at one’s own health (*n* = 110), the participants were asked to create a similar set of arguments in favor of vaccinating to protect their health (e.g., *If I ever get sick with COVID-19, it won’t be as dangerous as it would be if I weren’t vaccinated)*. Participants in both experimental groups were informed that their arguments would be rated by a group of experts to select the most persuasive ones. The participants then completed a set of measures, as described in the following section. All items used in the study, as well as the experimental manipulations, are available in the Files S1, S2 and S3, except for the empathy measure due to copyright.

#### 2.1.2. Measures

Besides the Empathic Sensitivity Scale [39], all the measures were developed based on Francis et al.’s [40] guideline on creating TPB questionnaires. Those items, to which participants responded using a 7-point Likert scale, were used in prior studies where they presented adequate reliability and validity [17,21]. The following measures were used in both Study 1 and Study 2:

**Empathy:** The 21-item Empathic Sensitiveness Scale [39] was used to measure the emotional and cognitive aspects of empathy. This measure is based on the Interpersonal Reactivity Index [27]. Davis postulated that empathy consists of several separate emotional and cognitive constructs, such as fantasy, perspective-taking, empathic concern, and personal distress. In the Polish version, only three dimensions (perspective-taking, empathic concern, and personal distress) have been distinguished. The items (e.g., *The suffering of others requires me to be compassionate and caring*) are scored on a 5-point Likert scale (1 = *completely disagree* to 5 = *completely agree*).

**Utility Beliefs:** Judgements about the possible consequences of being vaccinated against COVID-19 were assessed using self-created three seven-point bipolar adjective rating scales (e.g., “unprotective/protective”).

**Control Beliefs:** Beliefs about the degree of control over getting vaccinated against COVID-19 when it becomes available were assessed using three items (e.g., “It’s mostly up to me whether I get vaccinated against COVID-19 when a vaccine becomes available”) rated on self-created seven-point bipolar scales with varying anchors (i.e., completely disagree/agree, very hard/easy, no control at all/full control).

**Social Norm Beliefs:** Beliefs about the attitudes and opinions of others, especially those important to participants, about participant’s vaccination intention were assessed using three self-created items (e.g., “Most people who are important to me would praise me for getting vaccinated against COVID-19”) rated on a seven-point scale (1 = *completely disagree* to 7 = *completely agree*).

**Moral Norm Beliefs:** Beliefs about the participant’s moral responsibilities regarding COVID-19 vaccination were assessed using three self-created items (e.g., “Getting vaccinated against COVID-19 should be a moral obligation for all people”) rated on a seven-point scale (1 = *completely disagree* to 7 = *completely agree*).

**Vaccination Intention:** Intention to vaccinate against COVID-19 when a free vaccine becomes available was assessed using three self-created items (e.g., “When the vaccine becomes available, I will vaccinate myself”) rated on a seven-point scale (1 = *completely false* to 7 = *completely true*).

**Demographic Data:** The participants were asked to provide their age and sex, as well as whether they were afflicted with COVID-19 (S-COVID-19) or if they knew someone who was afflicted with COVID-19 (K-COVID-19).

### 2.2. Analysis

We conducted structural equation modeling (SEM) in R with the Lavaan library [41,42] using nonparametric statistics and estimators because of the lack of normality of the tested variables (K–S test; *p* < 0.001). Before the analyses, we prepared the data by (1) removing duplicated responses and responses with missing data; (2) coding sex as a binary dummy variable and excluding participants who did not report their sex or reported their sex as “Other/Non-binary,” due to an insufficient number of participants in this category; (3) coding the K-COVID-19 as a binary dummy variable by combining “no” and “don’t know” responses, owing to a small number of participants who reported not knowing whether an acquaintance was afflicted with COVID-19; (4) dummy coding the experimental group variable in such a way that there was a variable for each self-persuasion group compared to the direct persuasion group, to allow comparison of those groups in the following regressions in SEM (descriptives classified by both of those variables are reported in Appendix A). Finally, 14 responses were removed, and thus, 352 responses were included in further analyses.

Confirmatory factor analysis (CFA) and reliability analyses were conducted first. We used the weighted least squares mean (WLSM) estimation method, which performs well when using non-normally distributed and categorical data [43]. As Appendix A shows, the control beliefs scale had one item removed since its factor loading was lower than 0.3. The empathy scale was also reduced by removing an item and conducting another CFA after each removal until all items had factor loadings higher than 0.3. This resulted in an 11-item empathy scale that was used in further analyses. Next, we conducted reliability analyses, Spearman’s rho correlation analysis, and calculated the heterotrait-monotrait ratio of correlations (HTMT; [44]), which are reported in Table 2 and Table 3, respectively. Table 4 also contains Cronbach’s alpha reliabilities and average variances extracted (AVE), used to test for divergent validity of the measures used in the study. For this, we adopted two criteria. One is the Fornell–Larcker criterion [45], which assumes that AVE should not exceed the squared correlation for the divergence of the variables. To simplify the calculations, we used the square root of AVE instead. Two, HTMT should not exceed 0.85, or 0.90, if the measured constructs are similar. We adapted the former value for the empathy measure, while for the remaining variables, we adopted the latter. The moral norm beliefs measure was excluded from the analyses because they did not meet the HTMT criterion (HTMT_Moral norms-Intention_ = 0.94; HTMT_Moral norms-Utility beliefs_ = 0.90), as reported in Table 2.

Next, we performed SEM (WLSM). The initial model evaluating vaccination intention consisted of four observed exogenous variables (self-persuasion: own health; self-persuasion: others’ health; sex; K-COVID-19) and five endogenous latent variables (empathy, vaccination intention, control beliefs, utility beliefs, social norm beliefs, moral norm beliefs). K-COVID-19 was removed from the final model due to a lack of significance. We also estimated the error covariance among utility, social norms, and control beliefs, as well as between the self-persuasion variables. The factor loadings for every item included in the final model are presented in Appendix A, while the simplified model with only significant values is presented in Figure 1. We used standardized values.

### 2.3. Results

The developed model has good fit indices, indicating that it fits the data well. We used the robust versions of the following indices: Tucker–Lewis index (TLI), comparative fit index (CFI), root-mean-square of error of approximation (RMSEA), and standardized root-mean-square residual (SRMR), the values of which are reported in Table 3 alongside the suggested cutoff lines. Chi^2^ was significant even with the Satorra–Bentler correction (*chi*^2^ = 745.06, *df* = 278, *p* < 0.001).

As shown in the model, the experimental manipulation focused on protecting the health of others had a significant effect on utility beliefs (β = 0.18) and social norm beliefs (β = 0.15), while empathy had a significant effect on all TPB components (β_utility beliefs_ = 0.18, β_social beliefs_ = 0.16, β_control beliefs_ = 0.24). Simultaneously, being affected by sex, females were more likely to have reached high empathy values (β = −0.32). Furthermore, all TPB components had a significant effect on vaccination intention (β_utility beliefs_ = 0.46, β_social beliefs_ = 0.25, β_control beliefs_ = 0.27). There were also some noteworthy indirect effects. These included the effect of self-persuasion: others on vaccination intention through utility beliefs (β = 0.081), the effect of sex on control beliefs through empathy (β = −0.079), and the effect of empathy on vaccination intention through both utility (β = 0.080) and control beliefs (β = 0.066). These data are presented in Appendix A.

### 2.4. Discussion

As expected, the results of Study 1 demonstrated that among young adults, self-persuasion focused on others’ health (compared to self-persuasion targeted at one’s own health and direct persuasion) had an indirect effect on vaccination intention against COVID-19 through utility and social norm beliefs. As hypothesized, we found that as empathy increases, utility, social norms, and control beliefs increase. Greater empathy was associated with greater intention to vaccinate against COVID-19 through utility and control beliefs. Moreover, gender did not significantly influence the intention to vaccinate or TPB components. Women were more empathetic than men. Through their relationship with empathy, being female was associated with more positive attitudes (albeit almost significantly) toward vaccination and greater perceived behavioral control. Finally, knowing someone with COVID-19 was unrelated to the intention to vaccinate and all TPB components.

## 3. Study 2

A limitation of Study 1 was the difficulty of using self-persuasion to encourage young adults to be vaccinated against COVID-19 on a large scale. A method of inducing self-persuasion that overcomes these limitations is using open-ended questions that can effectively encourage specific health behaviors [48,49]. Prior studies on the effectiveness of posters on alcohol consumption demonstrated that messages framed as open-ended questions trigger self-generation of arguments for drinking less alcohol, which subsequently reduces actual alcohol consumption among young adults [50]. This study also showed that messages framed as open-ended questions evoked a less reactive state than direct persuasion, which may explain the effectiveness of self-persuasion in changing unhealthy behavior. The great advantage of messages framed as open-ended questions is that they are both effective and simple.

Similar to Study 1, we used self-persuasion targeted at protecting the health of others. In line with Study 1, we aimed to test whether components of the TPB model (utility, control, social, and moral norm beliefs) mediate between self-persuasion and intention to vaccinate and examine the effects of gender and knowing a person with COVID-19 on intention to vaccinate. Thus, by applying the results of Study 1, Study 2 investigated whether self-persuasion activated by posters with messages framed as open-ended questions, compared to direct persuasion, could effectively encourage vaccination against COVID-19 among young adults.

### 3.1. Participants, Materials, and Methods

In October 2020, 375 Polish adults participated in a separate online study hosted on Microsoft Forms. Of them, most were women (70.7%). No responses were removed. Similar to Study 1, the participants were recruited by disseminating study invitations on Facebook. All participants provided informed consent. The data used in the study are available at the OSF, while the specific demographics of the participants are presented in Table 5. There was no missing data.

Participants were randomly assigned to one of the two groups. In both groups, participants were asked to choose between two infographics (with the same message but different designs) that best persuaded them to take the vaccine. In addition, the participants were told that the questionnaire consisted of two different studies to create the impression that the manipulation and the later survey were not connected. Participants were told that the aim of the “first” study was to create a graphic that effectively persuades people to take the vaccine, while the aim of the “second” study was to explore factors leading to vaccination.

The direct persuasion group (*n* = 183) was presented with two images of young adults and an imperative sentence (“Protect yourself and your own health—take the coronavirus vaccine!”), which encouraged people to protect their health by getting vaccinated. In the focus on protecting others’ health group (*n* = 192), participants were presented with two similar images of older people, both accompanied by a short question (“How can you protect other people, especially those who cannot take the vaccine due to health problems, through vaccinating?”), which was intended to encourage them to take the vaccine to protect others’ health. Participants subsequently completed a short questionnaire comprising the same measures as in Study 1, except for the empathy questionnaire. The images used in the study were adapted from the Twitter account of the Polish Ministry of Health (https://twitter.com/MZ_GOV_PL/status/1243787403958947840 (accessed on 1 October 2020)) and the WHO website (https://www.who.int/bangladesh/emergencies/coronavirus-disease-(COVID-19)-update/advice-for-public---communicating-severity-series/images/default-source/searo---images/countries/bangladesh/infographics/risk-comms/english/slide5?itemIndex=5 (accessed on 1 October 2020)). They were edited so that the older or younger adults were present in the foreground of the image. Original images are available in the footnote, while the edited images used in the study are attached in the Files S1, S2 and S3. We conducted SEM using the same procedure as in Study 1. No participants were excluded.

### 3.2. Analysis

The CFA (Appendix A) indicated that two items belonging to the control scale had low factor loadings. Consequently, those items were removed. The divergent validity analyses (Table 6 and Table 7) indicated that the moral norm and utility belief measures did not meet the HTMT criterion (HTMT_Moral norms_—_Intention_ = 0.93; HTMT_Utility beliefs_—_Intention_ = 0.91).

The initial model for this study was comprised of three observed exogenous variables (sex, K-COVID-19, group) and three endogenous latent variables (control beliefs, social norm beliefs, vaccination intention). The K-COVID-19 variable was removed from the final model because of its lack of significance. The factor loadings are presented in Appendix A, and a simplified model with only significant values is presented in Figure 2. At the same time, descriptives of the variables classified by each group are reported in Appendix A. Standardized values have been reported.

### 3.3. Results

As reported in Table 8, the developed model had good fit indices, although Chi^2^ with the Satorra—Bentler correction was significant (*chi*^2^ = 60.081, *df* = 27, *p* < 0.001). Nevertheless, the model fits the data well. Results indicated that all variables had significant direct effects on vaccination intention (*β*_control beliefs_ = 0.54; *β*_social norm beliefs_ = 0.39; *β*_sex_ = −0.07; *β*_group_ = 0.27). These results indicated that women were more likely to have higher vaccination intention, similar to people who were assigned to the experimental group who focused on protecting the health of others. In accordance with the data presented in Appendix A, no significant indirect or total effects were detected.

### 3.4. Discussion

The results of Study 2 indicated the potential for a large-scale application of self-persuasion to encourage young adults to be vaccinated against COVID-19. We found that a poster with messages framed as open-ended questions that activated self-persuasion targeted at others’ health resulted in a greater intention to vaccinate against COVID-19 than direct persuasion. Because of the strong correlation between utility beliefs and intentions, we had to remove this variable from the model.

Thus, the model in Study 1 was fundamentally different from that in Study 2. In contrast to Study 1, wherein the effect of self-persuasion was mediated by attitude and norms, in Study 2, the effect of self-persuasion on increased intention to vaccinate was direct. In Study 2, gender affected intention directly—women were more likely to be vaccinated against COVID-19 than men. As in Study 1, the link between moral norms and intention in Study 2 was too strong to include moral norms in the model, while knowing someone with COVID-19 was not a significant predictor of intention to vaccinate.

## 4. General Discussion

Young adults could be encouraged to be vaccinated against COVID-19 by self-persuasion focused on others’ health. Self-persuasion focused on others’ health through presenting with arguments (Study 1) or being exposed to a poster with open-ended questions (Study 2), which resulted in greater vaccination intention than self-persuasion focused on one’s own health (Study 1) or direct persuasion (Studies 1 and 2). Thus, we demonstrated that self-persuasion consistently outperformed its direct persuasion counterpart. We also found that highly empathetic young adults have a higher intention to vaccinate because they have higher utility and control (Study 1).

Our findings demonstrated that among young adults, self-persuasion targeted at others’ health leads to higher vaccination intention against COVID-19 than self-persuasion targeted at one’s own health and direct persuasion. This finding is consistent with studies showing that self-persuasion can increase the intention to vaccinate against HPV [16]. Based on previous research findings, we explain the effectiveness of self-persuasion, compared to direct persuasion, as follows: (1) self-persuasion leads to less reactance than direct persuasion [12,13]; (2) self-generated arguments arouse the need to reduce cognitive dissonance by changing one’s attitude [14]; and (3) self-generated arguments were tailored to people’s concerns to a greater extent than arguments from direct persuasion [15].

Self-persuasion targeted at protecting one’s health was less effective in encouraging vaccination than self-persuasion targeted at protecting the health of others. Thus, it is crucial to tailor the arguments that activate self-persuasion focused on protecting others’ health. We argue that since COVID-19 does not pose a great risk to young adults’ health [23], they perceive it as less severe than older adults [21,22]. Therefore, self-generated arguments focused on vaccination for protecting personal health are less convincing for young adults than arguments focused on protecting others’ health. Previous studies have already shown that concern for others’ health is an important motivation for getting vaccinated [24,25]. Thus, our findings contribute to previous works that have conceptualized vaccination as a prosocial behavior [24,29,50].

Study 1 expands those works by showing that: (1) self-persuasion targeting prosocial aspects of vaccination can increase motivation to vaccinate; (2) motivation based on prosocial aspects of vaccination is stronger among young adults than motivation based on personal health benefits. Our findings showed that among young adults, self-persuasion targeted at protecting others’ health was more effective in encouraging vaccination than self-persuasion targeted at protecting one’s own health. This finding contradicts the results of Ashworth et al. [7]. Their study demonstrated that direct persuasion focused on personal health benefits is a more effective approach to enhancing COVID-19 vaccination intention than direct persuasion focused on others’ health benefits. However, this study was conducted on the general population, and therefore, the results cannot be generalized to young adults, who are less concerned about personal health than older adults [51]. Likewise, the results of our study on young adults cannot be generalized to the entire population.

The results of Study 2 indicated that self-persuasion could be used on a large scale. A poster with messages framed as open-ended questions effectively encouraged vaccination against COVID-19 among young adults compared to a poster containing direct persuasion. As in Study 1, we targeted self-persuasion to protect the health of others by presenting participants with the following question: “How can you protect other people, especially those who cannot take the vaccine due to health problems, through vaccinating?”. Based on previous studies [52], we believe that open-ended questions trigger self-generation of arguments for why vaccination against COVID-19 can protect others’ health, subsequently enhancing vaccination intention. Furthermore, we argue that our message framed as an open-ended question evoked a lower reactance state than direct persuasion, which may partly explain the effectiveness of self-persuasion targeted at protecting others’ health in promoting vaccination intention.

Since our findings show that self-persuasion is more effective than direct persuasion, large-scale application of our findings might be beneficial. Study 2′s findings suggest that self-persuasion can be successfully applied to media communications by framing health messages as open-ended questions. The great advantage of messages framed as open-ended questions is that they are both effective and simple; therefore, they could be applied in persuasive media messages encouraging COVID-19 vaccination. Self-persuasion by open-ended questions can be placed in various contexts: in health campaigns on television, radio, in the press, on the internet, in interactions between health education practitioners and young adults, or between doctors and patients. Self-persuasion seems to be effective only when people experience full freedom to choose their behavior [53]. Thus, practitioners should avoid pushing people too hard (through rewards or punishment) to create an argument for COVID-19 vaccination. Caution should also be exercised as self-persuasion receivers should know about vaccination to be able to generate arguments as to why vaccination can protect others’ health. Our study sample mostly included students who were likely to possess the knowledge necessary to create arguments. It seems that self-persuasion could be combined with education about vaccination for less-educated groups.

The current findings may have important implications for research on self-persuasion. The present work extends the classic work on self-persuasion by Aronson [11] and shows that self-persuasion can be strategically targeted to generate different arguments. The differences in the effectiveness of the two forms of targeted self-persuasion demonstrated in Study 1 suggested that targeting self-persuasion can be controlled. Thus, our results show the possibility of increasing the effectiveness of self-persuasion by addressing the content of generated arguments. Our findings open new avenues of research that can test what kinds of self-generated arguments can more effectively persuade people to engage in specific health behaviors. Targeted self-persuasion can be adopted for interventions concerned with other health behaviors, such as reducing smoking and alcohol consumption or promoting healthy eating.

The interpretation of our results must be contextualized to the pandemic conditions during the realization of the study, particularly the extent to which direct persuasion encouraging vaccination presented in the media by governments was used. The study was carried out just before the COVID-19 vaccines were introduced in Poland and before the government, doctors, and other authorities started to encourage COVID-19 vaccination. At that time, pressure measures to promote vaccination through, for example, vaccination certificates required to engage in various social activities were not yet used. Thus, self-persuasion was effective under conditions where most participants were unlikely to have experienced external pressure encouraging vaccination. Most likely, their reactance mechanisms toward these pressures were not aroused. It is not known whether self-persuasion would still be effective during an intensive campaign encouraging vaccination against COVID-19. Instead, we take the position that the results of our study suggest the use of self-persuasion at the initial stage of population vaccination before the use of direct persuasion.

### 4.1. TPB

By applying TPB, our results provide unique insights into the mechanism by which self-persuasion influences change in intention to vaccinate. First, consistent with previous findings [21,32], we demonstrated that utility, control, and social norm beliefs significantly predict the intention to vaccinate against COVID-19. Moral norms were found to be strongly associated with the intention to vaccinate to be included as a separate variable in the examined models. Such a strong relationship between the variables supports the idea that people perceive vaccinating against COVID-19 as a strong moral, prosocial behavior. Second, we found that self-persuasion targeted at others’ health (compared to self-persuasion targeted at one’s own health and direct persuasion) indirectly affects vaccination intention against COVID-19 through utility and social norms beliefs.

We argue that young adults under self-persuasion were targeted to protect others’ health created arguments by indicating that vaccination is effective in protecting the health of others, which increases their utility beliefs. An example of the arguments given by participants is: “The fact that someone has a strong immune system is a very selfish way of looking at a pandemic. It’s not about you; it’s about others, so you should be vaccinated, not just so that you do not get sick but so that other people around you do not get hurt”. The lack of effect of self-persuasion on control beliefs suggests that arguments indicating that vaccination is an easy way to contribute to the protection of others did not arise in response to targeted self-persuasion.

Furthermore, we consider that creating arguments indicating vaccination benefits for others reinforced the belief that other people expect one to vaccinate, which influenced the increase in their social norm beliefs. The identification in both studies of a link between social norms beliefs and intention to vaccinate supports previous findings showing that, for young adults, health behaviors are strongly affected by the behaviors of their peers [54]. Our results support existing literature concerning the effect of social norm beliefs on COVID-19 vaccination intention [55,56]. The strong need for acceptance and approval from peers that young adults feel makes them particularly susceptible to social norms [57]. This is why many young adults trust their friends’ COVID-19 opinions displayed on social media more than they trust the government, healthcare system, or scientists [58]. Therefore, since the behavior and attitudes of young adults’ peers can be such a powerful source of influence, government vaccination campaigns should use information about how many young adults have already been vaccinated against COVID-19—in particular, promoting a social media campaign where young people mark on their profiles that they have been vaccinated can be very effective.

Overall, the present findings contribute to the existing literature by identifying pathways from TPB mediating the effect of self-persuasion targeted at protecting the health of others on vaccination intention. This supports our understanding of the mechanism underlying the intervention.

### 4.2. Empathy

Self-persuasion focused on protecting others’ health can effectively increase young adults’ willingness to vaccinate, which corroborates our finding that as empathy increases, the intention to vaccinate increases. This finding is consistent with previous studies indicating that vaccinating oneself is a prosocial behavior; therefore, empathic concern motivates people to protect others’ health through vaccination [25,29]. Highly empathetic people have an other-directed focus of attention, which manifests in seeking information about the needs of others [26]. Thus, it is typical for such people to focus on protecting the health of others when making decisions to be vaccinated. The link between empathy and intention to vaccinate against COVID-19 among young adults supports our reasoning that an important factor in young adults’ motivation to be vaccinated is their desire to protect others. It is possible that self-persuasion focused on protecting the health of others also increases empathy levels, which may be one of the key mechanisms explaining the relationships observed in our study. Future studies should empirically test this possibility by measuring state empathy following the activation of self-persuasion focused on protecting others’ health.

Our findings provide a deeper understanding of the relationship between empathy and intention to vaccinate through the application of TPB. Thus, these findings extend those of previous studies [26] by showing that empathy is linked with a greater intention to vaccinate through a more positive attitude toward vaccination and increased perceived behavioral control. The positive association between empathy and TPB components may be due to the tendency of highly empathetic people to focus on others’ well-being [23]. This leads them to perceive the great benefits of vaccination in the interest of others and to judge that vaccination is an easy and low-effort way to help others.

### 4.3. Gender

Contrary to previous studies [21,59] and our hypothesis, men had no greater intention to be vaccinated than women. Not only have we found that women were more likely to get vaccinated against COVID-19, but we also confirmed that being a female was associated with more positive attitudes toward vaccination and greater perceived behavioral control [26], which was first related to women being more empathetic than men. The relationships between gender and the intention to vaccinate against COVID-19 appear to be complex and sample-specific. Possibly, we failed to replicate previous findings because gender differences in motivation to vaccinate were smaller among the young adult group.

### 4.4. Proximity with COVID-19

Contrary to previous studies [21,37] and our hypothesis, we found that among young adults, knowing someone afflicted with COVID-19 was not significantly correlated with the intention to vaccinate against COVID-19 and with TPB components. It may be that during the study period, young adults did not have as much direct contact with severe cases of COVID-19 as older people. It may be argued that more acquaintances of older people than young adults suffered from severe infection with COVID-19. Thus, limited contact with severely ill people with COVID-19 may have influenced the lack of a significant relationship between knowing someone afflicted with COVID-19 and the intention to vaccinate.

### 4.5. Limitations

Several limitations of this study need to be considered. First, our study participants were Facebook users, which may limit the generalizability of the results. Our sample was recruited through social media, which could have affected its bias. For example, the population we sampled mainly completed higher education. Further studies might test if the same models hold for young adults with lower education. Second, our study did not measure actual vaccination behavior because the timing of the studies preceded the widespread availability of COVID-19 vaccines. We measured the intention to vaccinate in the future. Notably, prior research [60] strongly indicates that the intention of a given behavior is a significant predictor of actual future behavior. Therefore, prospective studies should also incorporate measures of actual behavior. Third, in Study 2, the control beliefs scale consisted of only two items, while it is a good practice to have at least three items per scale. This also affected the reliability of the scale. In future studies, this scale will be revised. Finally, we conducted our studies when vaccination was not yet available in Europe but only heavily debated. Thus, self-persuasion increased participants’ intention to vaccinate when attitudes toward COVID-19 vaccination were not clearly defined among a significant number of people. Future studies should investigate whether self-persuasion will be effective at present times when COVID-19 vaccines have been made available, and attitudes toward COVID-19 vaccination are clearer.

## 5. Conclusions

Our findings provide a deeper understanding of the interplay between self-persuasion and factors that motivate young adults to be vaccinated against COVID-19. Our results extend the existing knowledge of the relationship between self-persuasion and vaccination intention and suggest that among young adults, self-persuasion targeted at protecting others’ health, compared to direct persuasion, can affect declared vaccination intention through utility and social norm beliefs. Our results provide insight into “why” people are motivated to vaccinate against COVID-19 and accordingly contribute to the scientific pursuit of encouraging young adults to get vaccinations. Finally, our research has not only theoretical but also practical consequences, as knowing that self-persuasion can affect declared intention towards vaccination can help to effectively encourage young adults to be vaccinated against COVID-19.

Encouraging young adults to be vaccinated is difficult, as they do not perceive COVID-19 as a severe disease to the extent older people do [21,22]. Thus, evidence-based guidelines are needed to effectively motivate young adults. Taken together, the findings contribute to the list of evidence-based methods of encouraging vaccination against COVID-19. The findings, thus, have important implications for ongoing governmental interventions via mass media designed to encourage vaccination among young adults.

## Figures and Tables

**Figure 1 vaccines-10-00553-f001:**
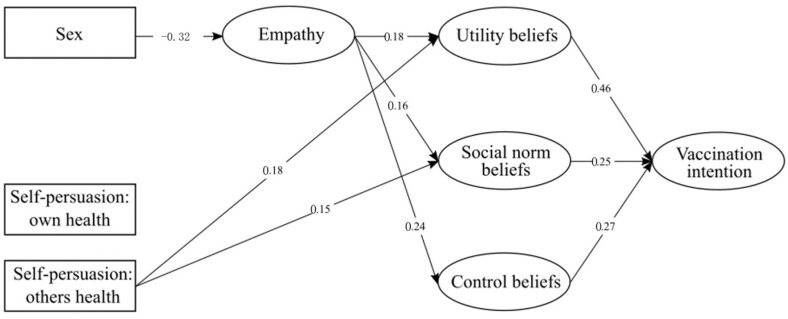
Simplified SEM model for Study 1 with only significant values.

**Figure 2 vaccines-10-00553-f002:**
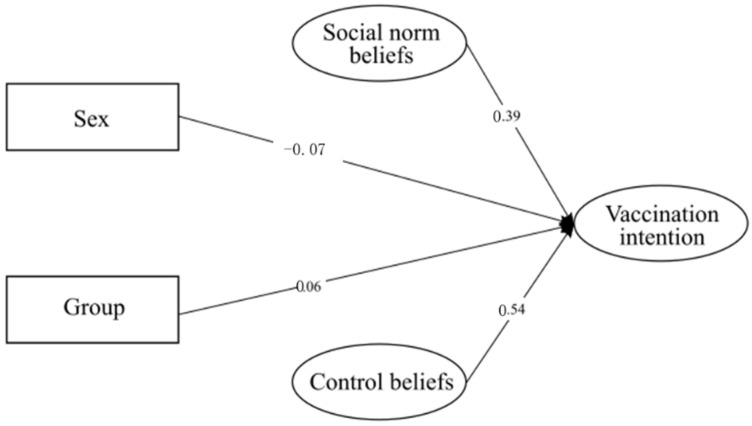
Simplified SEM model for Study 2 with only significant values.

**Table 1 vaccines-10-00553-t001:** Participants’ demographic data (Study 1).

Descriptive Variable	*n*	%
Age (*M*; *SD*)	*M* = 22.37; *SD* = 2.23	
18–20	67	19
21–25	252	71.5
26–30	33	9.5
Sex		
Male	64	18.2
Female	280	79.5
Other/Non-binary	5	1.4
No data	3	0.9
Have you gotten sick with COVID-19?		
No	203	57.5
Yes	3	0.9
I don’t know	146	41.5
Do you personally know someone who has gotten sick with COVID-19?		
No	165	46.9
Yes	136	38.6
I don’t know	51	14.5

**Table 2 vaccines-10-00553-t002:** Heterotrait-monotrait ratio of correlations for Study 1.

	1	2	3	4	5	6
*1. Empathy*	-					
*2. Intention*	0.17	-				
*3. Control beliefs*	0.25	0.86	-			
*4. Utility beliefs*	0.15	0.88	0.86	-		
*5. Social norm beliefs*	0.14	0.81	0.77	0.79	-	
*6. Moral norm beliefs*	0.21	0.94	0.85	0.90	0.84	-

**Table 3 vaccines-10-00553-t003:** Fit measures for Study 1.

Measure	Estimate	Cutoff [46,47]
Tucker–Lewis Index (TLI)	0.969	>0.95
Comparative Fit Index (CFI)	0.973	>0.95
Root-Mean-Square Error of Approximation (RMSEA)	0.051	<0.08
Standardized Root-Mean-Square Residual (SRMR)	0.060	<0.08

**Table 4 vaccines-10-00553-t004:** Spearman’s rho, means, standard deviations, and average variance were extracted in Study 1.

	1	2	3	4	5	6	7	8
*1. Empathy*	-							
*2. Intention*	0.13 *	-						
*3. Utility beliefs*	0.12 *	0.77 **	-					
*4. Control beliefs*	0.17 **	0.72 **	0.68 **	-				
*5. Social norm beliefs*	0.09	0.71 **	0.69 **	0.63 **	-			
*6. Moral norm beliefs*	0.13 *	0.85 **	0.77 **	0.69 **	0.75 **	-		
*7. Sex*	−0.25 **	0.03	0.08	−0.01	0.06	0.03	-	
*8. K-COVID-19*	0.06	0.15 **	0.16 **	0.12 *	0.18 **	0.12 *	0.09	-
*M*	3.40	5.20	5.38	5.17	4.76	5.01	-	-
*SD*	0.68	1.95	1.57	1.50	1.83	1.89	-	-
*α*	0.81	0.96	0.92	0.77	0.96	0.94	-	-
*AVE*	0.31	0.90	0.80	0.61	0.88	0.84	-	-
*√AVE*	0.56	0.95	0.89	0.78	0.94	0.91	-	-

Note * *p* < 0.05, ** *p* < 0.01; sex was dummy coded as 0 = Female, 1 = Male, while K-COVID-19 as 0 = No, 1 = Yes. K-COVID-19—knowing someone afflicted with COVID-19.

**Table 5 vaccines-10-00553-t005:** Participants’ demographic data (Study 2).

Descriptive Variable	*n*	%
Age (*M*; *SD*)	*M* = 21.07; *SD* = 2.19	
18–20	182	48.5
21–25	178	47.5
26–30	15	4
Sex		
Male	106	28.3
Female	265	70.7
Other/Non-binary	4	1.1
No data	0	0
Have you gotten sick with COVID-19?		
No	215	57.3
Yes	3	.8
I don’t know	157	41.9
Do you personally know someone who has gotten sick with COVID-19?		
No	162	43.3
Yes	171	45.6
I don’t know	42	11.2

**Table 6 vaccines-10-00553-t006:** Spearman’s rho, means, standard deviations, and average variance extracted in Study 2.

		1	2	3	4	5	6	7
*1. Intention*		-						
*2. Utility beliefs*		0.80 **	-					
*3. Control beliefs*		0.35 **	0.33 **	-				
*4. Social norm beliefs*		0.65 **	0.65 **	0.24 **	-			
*5. Moral norm beliefs*		0.84 **	0.80 **	0.32 **	0.73 **	-		
*6. Sex*		0.03	0.11 *	−0.00	0.08	0.06	-	
*7.* *K-COVID-19*		0.09	0.10	−0.00	0.10	0.12 *	−0.02	-
*M*		5.08	5.30	5.59	4.47	4.97	-	-
*SD*		1.92	1.63	1.04	1.77	1.90	-	-
*α*		0.95	0.92	0.81	0.95	0.95	-	-
*AVE*		0.88	0.82	0.38	0.87	0.87	-	-
*√AVE*		0.93	0.90	0.62	0.93	0.93	-	-

Note. * *p* < 0.05, ** *p* < 0.01, sex was dummy coded as 0 = Female, 1 = Male, while K-COVID-19 as 0 = No, 1 = Yes.

**Table 7 vaccines-10-00553-t007:** Heterotrait-monotrait ratio of correlations for Study 2.

	1	2	3	4	5
*1. Intention*	-				
*2. Control beliefs*	0.78	-			
*3. Utility beliefs*	0.91	0.73	-		
*4. Social norm beliefs*	0.73	0.63	0.72	-	
*5. Moral norm beliefs*	0.93	0.78	0.89	0.80	-

**Table 8 vaccines-10-00553-t008:** Fit measures for Study 2.

Measure	Estimate	Cutoff [46,47]
Tucker–Lewis Index (TLI)	0.998	>0.95
Comparative Fit Index (CFI)	0.999	>0.95
Root-Mean-Square Error of Approximation (RMSEA)	0.019	<0.08
Standardized Root-Mean-Square Residual (SRMR)	0.018	<0.08

## Data Availability

The dataset is available for free download from the Open Science Framework (https://osf.io/ckf2v/?view_only=42799b0d579b437c8cb926ccfddbfba0, accessed on 1 March 2021).

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
