# Peer review of "Vaccinating to Protect Others: The Role of Self-Persuasion and Empathy among Young Adults"

_vaccines, 2022, doi:10.3390/vaccines10040553_

Round 1
Reviewer 1 Report
The manuscript provides an interesting insight into the role of self-persuasion and empathy among young adults in increasing vaccination intention against COVID-19.
Analysis has been performed correctly, a few comments to improve the readability of the paper.
Introduction
I would restructure the introduction to make it one comprehensive section, without subheadings and with no distinction for study 1 and 2.
In addition, other studies have investigated the reasons for vaccination uptake among university students, including DOI: 10.3390/vaccines9060585, DOI: 10.3390/vaccines9080927, and DOI: 10.3390/vaccines9111292). These should be added.
Methods
Why are table 1, 2 and 3 included in the methods section?
Probably would be better to not say Study 1 or 2 but Group 1 or 2, since it seems that the study was one with different questions for participants. This applies to the Results section as well.
How were the 7-point scales chosen?
Discussion
I would not divide it into Study 1 and 2 (or Group 1 and 2) since it should be a comprehensive section discussing the overall findings of your study.
Please be careful not to repeat results in this section (for example, line 496-449 etc.).
Author Response
Comment 1: The manuscript provides an interesting insight into the role of self-persuasion and empathy among young adults in increasing vaccination intention against COVID-19. Analysis has been performed correctly, a few comments to improve the readability of the paper.
Answer 1: Thank you!
Comment 2: I would restructure the introduction to make it one comprehensive section, without subheadings and with no distinction for Studies 1 and 2.
Answer 2: As suggested by the reviewer, we have removed the subheadings from the Introduction. However, we chose not to combine the description of Studies 1 and 2 since they are separate studies, contrary to the suggestion in comment 5. The assumptions and aims of Study 2 follow directly from the conclusions made based on Study 1, for example, we used only self-persuasion targeted at protecting the health of others in Study 2, as self-persuasion targeted at protecting one's own health was ineffective in Study 1.
Comment 3: In addition, other studies have investigated the reasons for vaccination uptake among university students, including DOI: 10.3390/vaccines9060585, DOI: 10.3390/vaccines9080927, and DOI: 10.3390/vaccines9111292). These should be added.
Answer 2: As suggested, we described the motivation of young people to be vaccinated against COVID-19 on the basis of the indicated three papers as follows:
Page 2: “To date, a number of motivational barriers to vaccination against COVID-19 have been identified among young people, including belief in conspiracy theories [18], and lack of confidence in the safety, authenticity, and efficacy of vaccination [19-20].”
Comment 4: Why are table 1, 2, and 3 included in the methods section?
Answer 4: Table 1 includes the characteristics of the participants and those, according to the APA Publication Manual (2020, 7th edition), belong in the Methods section. However, to make this clearer, we have changed the section heading to: “Materials, Participants and Methods.” Tables 2 and 3 are assigned to subsection “Analysis” of Study 1, which was mistakenly numbered as belonging to the “Methods” subsection. This was corrected for both studies - now “Analysis” is independent of “Methods.”
Comment 5: Probably would be better to not say Study 1 or 2 but Group 1 or 2, since it seems that the study was one with different questions for participants. This applies to the Results section as well.
Answer 5: Studies 1 and 2 are two completely separate studies - they were conducted at different times, on different groups of participants, using different tools. We, therefore, choose not to follow the suggested changes.
Comment 6: How were the 7-point scales chosen?
Answer 6: We have used those same scales in previous studies in such forms, where we have decided on 7-point scales. Those decisions were based on papers claiming (e.g., Tarka, 2017) that 7-point scales might behave similarly to continuous data in structural equation modeling. That, in turn, widens the range of potential estimators that we could use by the maximum likelihood estimators (e.g., ML, MLM, MLR) that are not a good fit for categorical data. In the final analysis, we did not use any of those estimators due to lack of data normality, requiring us to use one of the weighted least squares estimators better suited for non-normal data than any ML estimator. In addition to those statistical considerations, as Francis et al. (2004) indicate, 7-point scales are the most recommended response format in the Theory of Planned Behavior literature (e.g., Courneya, Conner, & Rhodes, 2005).
To further justify the use of a 7-point scale we have added the following sentence to the text:
Page 5: “Those items, to which participants responded using a 7-point Likert scale, were used in prior studies, where they presented adequate reliability and validity [17,18]”
Comment 7: I would not divide it into Study 1 and 2 (or Group 1 and 2) since it should be a comprehensive section discussing the overall findings of your study.
Answer 7: Section “Study 1” and section “Study 2” were merged as per your suggestion.
Comment 8: Please be careful not to repeat results in this section (for example, line 496-449 etc.).
Answer 8: We have corrected several sentences in the Discussion section to not sound like simple repetitions of the results. For instance, the sentence from the example you gave was changed to: “Not only have we found that women were more likely to get vaccinated against COVID-19, but also confirmed that being a female was associated with more positive attitudes toward vaccination and greater perceived behavioral control [23], which was firstly related with women being more empathetic than men.” We have ensured that no phrases or sentences were repeated in the Results and Discussion sections.
Reviewer 2 Report
Dear Authors,
I have perused with interest the article titled Vaccinating to protect others: The role of self-persuasion and empathy among young adults, in which you have attempted to lay out an in-depth analysis in one of the underlying determinants on which vaccination uptake is grounded: persuasion and empathy as motivating forces. Discussing self-persuasion and the psychological dynamics triggered over the whole process constitutes a strength of this article: originality in elucidating a somewhat underresearched aspect of the COVID-19 pandemic. Another strength is arguably its relevance, given how the pandemic is not at all behind us, and we as healthcare operators need to tweak our strategic approaches, learn from past mistakes and enhance our level of preparedness in order to tackle future extreme challenges.
I feel the abstract ought to be fine-tuned a bit and made more directly meaningful in terms of anticipating the contents and structure of the article. Given how you have chosen to mold your manuscript, I would advise you to resort to a structured abstract (Background/Objective, Methods, Results, Conclusions) elaborating in greater detail how the article is set to unfold. Please describe the aim more directly and overtly, then the methodology and so forth.
As for the article itself, I view it as a praiseworthy contribution that digs deeper into key complexities, and its various limitations (tapping into social media, timing prior to the widespread availability of vaccines and the like) have been adequately covered by the authors. The methodology appears sound insofar as I was able to determine. Figures and tables are effective enough at conveying key info. Still, when it comes to weaknesses, I believe the article falls short in terms of exploring the force and influence of other persuasion agents, and contextualize them within their own research findings. Namely, in light of the relatively young age of the pool surveyed, I advise you to expound upon the role played by peer-to-peer influence, likely the most consequentially impactful factor when it comes to changing behaviours. In outlining vaccination campaigns, many governments have acknowledged how behaviours and choices spread by social emulation are key: our peers (whether that be friends, colleagues, people in our proximity network) are a prominently powerful source of influence. While people may put some trust in the messages from public health officials, scientists or politicians, adherence will be substantially enhanced if they see and hear ‘people like themselves’ (peers, mates, colleagues) voicing their agreement or will to accept that advice. Exploring those dynamics may contribute to making the article more comprehensive. Furthermore, I would expound upon the power of (self-) persuasion against the backdrop of the various policies based on coercion or incentivization. How do those elements mesh and how do they affect vaccine hesitancy? Is one more or less likely to "self-persuade" if an element of coercion comes into the picture, e.g. vaccination certificates required to engage in various social activities? Certainly I would assume the effectiveness and extent of self-persuasion can be influenced by external elements and information. Your discussion needs to contextualize your findings within the framework of the pandemic scenario, with the numerous variables that come to play a role.
Line 573: "The results of previous research suggest that self-persuasion may be a more effective form of convincing than direct persuasion in countries with an individualistic culture than in countries with a collectivistic culture. In contrast, targeting persuasion to protect the health of others through vaccination may be less effective in countries with individualistic cultures than in collectivistic cultures".
These are broad-ranging affirmations that need much more thorough an analysis. Consider removing them from the manuscript if you do not want to put them into context and expound upon them much more extensively.
Overall, the article is a potentially valuable and insightful contribution in an all too important area of research. It needs a little tweaking before it can be greenlighted for publication though.
Although it is well-written overall, I would have it proofread again by a native speaker of English too, for the sake of greater clarity and better readability.
Author Response
Comment 1: Dear Authors,
I have perused with interest the article titled Vaccinating to protect others: The role of self-persuasion and empathy among young adults, in which you have attempted to lay out an in-depth analysis in one of the underlying determinants on which vaccination uptake is grounded: persuasion and empathy as motivating forces. Discussing self-persuasion and the psychological dynamics triggered over the whole process constitutes a strength of this article: originality in elucidating a somewhat underresearched aspect of the COVID-19 pandemic. Another strength is arguably its relevance, given how the pandemic is not at all behind us, and we as healthcare operators need to tweak our strategic approaches, learn from past mistakes and enhance our level of preparedness in order to tackle future extreme challenges.
Answer 1: Thank you for reading our paper carefully and pointing out its strengths.
Comment 2: I feel the abstract ought to be fine-tuned a bit and made more directly meaningful in terms of anticipating the contents and structure of the article. Given how you have chosen to mold your manuscript, I would advise you to resort to a structured abstract (Background/Objective, Methods, Results, Conclusions) elaborating in greater detail how the article is set to unfold. Please describe the aim more directly and overtly, then the methodology and so forth.
Answer 2: As suggested, we have divided the Abstract into four sections (Background, Methods, Results, and Conclusions) but without adding the actual headings, as this would be against the journal’s guidelines. We have also described the aim of our paper more directly. Regarding the methods, we added information about the use of tools derived from the theory of planned behavior and the use of SEM analyses. Finally, the entire Abstract was revised to fit within the 200-word limit.
Comment 3: As for the article itself, I view it as a praiseworthy contribution that digs deeper into key complexities, and its various limitations (tapping into social media, timing prior to the widespread availability of vaccines and the like) have been adequately covered by the authors. The methodology appears sound insofar as I was able to determine. Figures and tables are effective enough at conveying key info.
Answer 3: Thank you for listing the strengths of our paper.
Comment 4: Still, when it comes to weaknesses, I believe the article falls short in terms of exploring the force and influence of other persuasion agents, and contextualize them within their own research findings. Namely, in light of the relatively young age of the pool surveyed, I advise you to expound upon the role played by peer-to-peer influence, likely the most consequentially impactful factor when it comes to changing behaviours. In outlining vaccination campaigns, many governments have acknowledged how behaviours and choices spread by social emulation are key: our peers (whether that be friends, colleagues, people in our proximity network) are a prominently powerful source of influence. While people may put some trust in the messages from public health officials, scientists or politicians, adherence will be substantially enhanced if they see and hear ‘people like themselves’ (peers, mates, colleagues) voicing their agreement or will to accept that advice. Exploring those dynamics may contribute to making the article more comprehensive.
Answer 4: As suggested, we significantly expanded the discussion of the impact of social norms on COVID-19 vaccination among young adults. We have introduced the following sentences:
Page 14: “The identification in both studies of a link between social norms beliefs and intention to vaccinate supports previous findings showing that, for young adults, health behaviors are strongly affected by the behaviors of their peers [54]. Our results support existing literature concerning the effect of social norm beliefs on COVID-19 vaccination intention [55,56]. The strong need for acceptance and approval from peers that young adults feel makes them particularly susceptible to social norms [57]. This is why many young adults trust their friends' COVID-19 opinions displayed on social media more than they trust the government, healthcare system, or scientists [58]. Therefore, since the behavior and attitudes of young adults' peers can be such a powerful source of influence, government vaccination campaigns should use information about how many young adults have already been vaccinated against COVID-19—in particular, promoting a social media campaign where young people mark on their profiles that they have been vaccinated can be very effective.”
Comment 5: Furthermore, I would expound upon the power of (self-) persuasion against the backdrop of the various policies based on coercion or incentivization. How do those elements mesh and how do they affect vaccine hesitancy? Is one more or less likely to "self-persuade" if an element of coercion comes into the picture, e.g. vaccination certificates required to engage in various social activities? Certainly I would assume the effectiveness and extent of self-persuasion can be influenced by external elements and information. Your discussion needs to contextualize your findings within the framework of the pandemic scenario, with the numerous variables that come to play a role.
Answer 5: As suggested by the reviewer, we have introduced an additional paragraph to the Discussion in which we contextualize the results of our study in the context of the current pandemic stage.
Page 14: “The interpretation of our results must be contextualized to the pandemic conditions during the realization of the study, particularly the extent to which direct persuasion encouraging vaccination presented in the media by governments was used. The study was carried out just before just before the COVID-19 vaccines were introduced in Poland, and before the government, doctors, and other authorities started to encourage COVID-19 vaccination. At that time, pressure measures to promote vaccination through, for example , vaccination certificates required to engage in various social activities, were not yet used. Thus, self-persuasion was effective under conditions where most participants were unlikely to have experienced external pressure encouraging vaccination. Most likely, their reactance mechanisms toward these pressures were not aroused. It is not known whether self-persuasion would still be effective during an intensive campaign encouraging vaccination against COVID-19. Instead, we take the position that the results of our study suggest the use of self-persuasion at the initial stage of population vaccination, before the use of direct persuasion.”
Comment 6: Line 573: "The results of previous research suggest that self-persuasion may be a more effective form of convincing than direct persuasion in countries with an individualistic culture than in countries with a collectivistic culture. In contrast, targeting persuasion to protect the health of others through vaccination may be less effective in countries with individualistic cultures than in collectivistic cultures".
These are broad-ranging affirmations that need much more thorough analysis. Consider removing them from the manuscript if you do not want to put them into context and expound upon them much more extensively.
Answer 6: We agree with this comment and have therefore removed the indicated fragment from the manuscript as suggested.
Comment 7: Overall, the article is a potentially valuable and insightful contribution in an all too important area of research. It needs a little tweaking before it can be greenlighted for publication though.
Answer 7: Thank you once again for appreciating our work.
Comment 8: Although it is well-written overall, I would have it proofread again by a native speaker of English too, for the sake of greater clarity and better readability.
Answer 8: The revised manuscript was edited by professional proofreading company, Editage before resubmission.
Round 2
Reviewer 1 Report
The authors addressed all comments.
Reviewer 2 Report
Dear Authors,
I appreciate the way in which you have improved your manuscript, which is now more balanced, thorough and well-rounded overall.
You have provided answers addressing my remarks and suggestions and I am satisfied with that.
A psychological perspective and well elaborated behavioral science dynamics make the article original and relevant.
I would recommend that the article be deemed worthy of publication.
All the best.